# Signal Deconvolution and Noise Factor Analysis Based on a Combination of Time–Frequency Analysis and Probabilistic Sparse Matrix Factorization

**DOI:** 10.3390/ijms21082978

**Published:** 2020-04-23

**Authors:** Shunji Yamada, Atsushi Kurotani, Eisuke Chikayama, Jun Kikuchi

**Affiliations:** 1Graduate School of Bioagricultural Sciences, Nagoya University, Furo-cho, Nagoya 464-8601, Chikusa-ku, Japan; shunji.yamada@riken.jp; 2RIKEN Center for Sustainable Resource Science, 1-7-22 Suehiro-cho, Yokohama 230-0045, Tsurumi-ku, Japan; atsushi.kurotani@riken.jp (A.K.); chikaya@nuis.ac.jp (E.C.); 3Department of Information Systems, Niigata University of International and Information Studies, 3-1-1 Mizukino, Niigata 950-2292, Nishi-ku, Japan; 4Graduate School of Medical Life Science, Yokohama City University, 1-7-29 Suehiro-cho, Yokohama 230-0045, Tsurumi-ku, Japan

**Keywords:** NMR, molecular complexity, FID, short-time Fourier transform, matrix factorization, *T*_2_* relaxation time, diffusion-edited spectrum, signal-to-noise ratio, acquisition parameters, correlation network analysis

## Abstract

Nuclear magnetic resonance (NMR) spectroscopy is commonly used to characterize molecular complexity because it produces informative atomic-resolution data on the chemical structure and molecular mobility of samples non-invasively by means of various acquisition parameters and pulse programs. However, analyzing the accumulated NMR data of mixtures is challenging due to noise and signal overlap. Therefore, data-cleansing steps, such as quality checking, noise reduction, and signal deconvolution, are important processes before spectrum analysis. Here, we have developed an NMR measurement informatics tool for data cleansing that combines short-time Fourier transform (STFT; a time–frequency analytical method) and probabilistic sparse matrix factorization (PSMF) for signal deconvolution and noise factor analysis. Our tool can be applied to the original free induction decay (FID) signals of a one-dimensional NMR spectrum. We show that the signal deconvolution method reduces the noise of FID signals, increasing the signal-to-noise ratio (SNR) about tenfold, and its application to diffusion-edited spectra allows signals of macromolecules and unsuppressed small molecules to be separated by the length of the *T*_2_* relaxation time. Noise factor analysis of NMR datasets identified correlations between SNR and acquisition parameters, identifying major experimental factors that can lower SNR.

## 1. Introduction

NMR spectroscopy is one of the most powerful tools available for molecular characterization at the atomic level [1]. Because it is non-invasive, NMR has been applied to data-driven analyses of molecular complexity in many areas of health [2], food [3], materials [4], and the environment [5]. In measuring NMR signals, the main challenges are the sensitivity and resolution of the NMR spectrum [6]. On the one hand, various techniques and devices for improving sensitivity have been developed, such as high-field magnets [7], cryogenic detection systems [8], shimming and locking to adjust the magnetic field [9], and dynamic nuclear polarization [10]. In addition, pulsed field gradient (PFG), nonuniform sampling [11] and magnetization transfer techniques such as cross-polarization [12] and INEPT (Insensitive nuclei enhanced by polarization transfer) [13] have been developed to enhance the sensitivity per unit time. On the other hand, compact and benchtop NMR instruments with lower resolution have become highly cost-effective owing to marked progress in the materials used for the permanent magnet [14].

Regarding spectral resolution, many pulse sequences for the measurement of one-dimensional (1D)-NMR with selective signal suppression, including pre-saturation, Carr–Purcell–Meiboom–Gill (CPMG) [15], WATER suppression by GrAdient Tailored Excitation (WATERGATE) [16], diffusion-editing [17], double quantum filter [18], and pure shift NMR [19], have been developed to reduce signal overlap. However, the spectra have remaining overlapping signals, or the overlapping peaks themselves contain part of the information of the sample. In this regard, overlapping signals can be separated by two-dimensional (2D)-NMR, in which multiple free induction decays (FIDs) are measured over a small change in evolution time, but this approach is time consuming [20].

Conventionally, methods for improving the sensitivity and resolution of FIDs are adjusted by pre-processing steps, such as zero filling and apodization, before Fourier transformation (FT) is carried out [21]. Other methods for reducing mathematical noise from FID signals focus on the region of interest (ROI), such as reference deconvolution [22], harmonic inversion noise removal (HINR) [23], and complete reduction to amplitude frequency table (CRAFT) [24]. In addition, STFT and wavelet transform [25] have been developed as alternative transformation methods to FT for analyzing the relationship between the time and frequency of FIDs. In principle, the exponential decay constant of the FID obtained by applying a 90° pulse to create transverse magnetization is the *T*_2_ relaxation time, a physical parameter independent of field inhomogeneity. In reality, however, because of the effect of magnetic field inhomogeneity, the decay constant of the FID is defined as *T*_2_*, an instrument-dependent parameter, rather than *T*_2_. STFT has the ability to extract time-varying behavior from FIDs, allowing for the analysis of dynamic chemical shifts of atoms in flexible proteins [26]. In addition, it has been reported that STFT can extract *T*_2_* information from FIDs and improve the results of discriminant analysis [27]. Applying the same idea to covariance NMR [28], *T*_2_*-weighted covariance NMR improves the sensitivity and resolution of signals based on the difference in *T*_2_*, determined by dividing each FID in the *t*_1_ dimension of 2D-NMR to create a series of sub-FIDs [29]. In an alternative approach, matrix factorization (MF) is commonly used to extract signal components and separate peaks in spectra [30]. For example, a noise reduction method using principal component analysis (PCA), which is one of the most commonly used multivariate analysis methods for extracting features of data, has been applied to solid CP-MAS NMR data measured by various parameters [31]. Therefore, the quality and amount of information from FIDs can be maximized by applying corrections based on different characteristics. Nevertheless, all these methods require multiple FIDs obtained by adding either spectral dimensions or multiple conditions of samples or parameters. There is also a computational approach such as CORE (COmponent-REsolved; a multi-component spectral separation approach previously introduced method). It focuses on diffusion coefficients to separate the NMR signals of different compounds in PFG-NMR [32,33,34]. However, this technique requires a specific NMR probe with a coil for generating PFG.

In the current move toward a digital innovation society, tools for NMR measurement informatics are becoming increasingly important [35]. Alongside this, the value of raw NMR datasets for reuse in research studies is rising [36]. Although the quality of raw data influences the value of knowledge obtained in terms of both insight and prediction [37], data cleansing methods for utilizing various kinds of NMR data accumulated over many years, such as data quality checks, noise reduction, and signal deconvolution, have not been established.

In this study, by focusing on acquisition parameters [38,39,40,41,42] and noise [25], we have developed an NMR measurement informatics tool for data cleansing based on FID signal deconvolution and noise factor analysis. Our method for deconvoluting signals and noise factor analysis can be applied to original single FIDs from 1D-NMR and is based on STFT [43] and PMSF [44]. It differs from conventional noise reduction using multivariate analysis [34] because it does not require multiple 1D-NMR data that are measured on many samples or acquired with several acquisition parameters. The difference in *T*_2_* on the time axis determined by performing STFT for each frequency component is useful to separate signals based on MF instead of ROI [22,23,24]. Our method that focuses on the relaxation time utilizes the attenuation behavior of the FID signal without any hardware upgrade for NMR research field. Lastly, we have developed a function for collecting acquisition parameters as a measurement of experimental factors from a directory of NMR data, and investigated the relationship between signal-to-noise ratio (SNR) and acquisition parameters. A researcher performing NMR must select parameters for each experiment, and normally chooses a reasonable set of parameters based on their experience. We show that these parameters can be characterized in terms of their correlation with SNR by a statistical analysis of accumulated NMR datasets. Therefore, this method will be useful to determine the optimal conditions of acquisition parameters.

## 2. Results and Discussion

### 2.1. Signal Deconvolution Method

In this study, signal deconvolution, based on the combined method of STFT and PSMF, was applied to FIDs of 1D-NMR to separate the components and improve SNR. The theory behind the signal deconvolution method is described in detail in the Appendix A. In brief, in FT NMR spectroscopy, the FID is the NMR signal generated by non-equilibrium nuclear spin magnetization precessing along the magnetic field. In general, this non-equilibrium magnetization can be generated by applying a pulse of resonant radiofrequency close to the Larmor frequency of the nuclear spins of the sample. Each FID is commonly a sum of multiple decayed oscillatory signals. These signals return to equilibrium at different rates or relaxation time constants. Thus, analysis of the relaxation times of an FID for a sample gives significant insight into the chemical composition, structure, and mobility of the sample. FIDs acquired by NMR measurement are composed of many signals derived from the sample, in addition to several types of noise, such as external noise, physical vibration, power supply, and internal noise from the spectrometer due to thermal noise. Therefore, an FID can be modeled as:(1)$$S\left(t\right)=S_{signal}\left(t\right)+S_{noise}\left(t\right)$$
where *S*(*t*) is the measured signal, and *S_signal_*(*t*) and *S_noise_*(*t*) are sets of ideal signals and signals from different types of noise, respectively (Equation (1) and Appendix A) [45]. The relaxation process can then be described as the exponential decay of the transverse magnetization S(t) (Appendix A) [46]. The shorter the relaxation time *T*_2_*, the more rapid the decay. If an FID has more than one component, it will be the sum of contributions from each component (Appendix A).

Whereas standard FT (Appendix A) contains only the frequency domain, STFT contains both frequency and time domains. Because the FID signal decays exponentially with time, for STFT, it needs to be divided into several small time intervals (segments) to analyze the time–frequency feature accurately, and FT is used to determine the frequency feature of each segment, thereby increasing the accuracy of signal feature extraction. STFT uses a window function to obtain each weighted segment on the time axis, and then applies FT to each segment. STFT of *S*(*t*) can be written as:(2)$$STFT_{S}\left(\tau ,\omega \right)=\int_{-\infty }^{\infty }S\left(t\right)g\left(t-\tau \right)\exp\left(-i\omega t\right)dt$$
where the window function g is first used to intercept the progress of FT on *S*(*t*) around t=τ locally, and then FT of the segment is performed on *t* (Equation (2) and Appendix A) [43]. By moving the center position of the window function g sequentially, all the FTs at different times can be obtained. 

$STFT_{S}\left(\tau ,\omega \right)$ is a complex-valued function (Appendix A) composed of two types of signal: real (*Re*, Appendix A) and imaginary (*Im*, Appendix A), whose phases differ from each other by 90° (Appendix A). To change the complex value into an absolute value, the following equation is applied:(3)$$\left|z\right|=\sqrt{Re^{2}+Im^{2}}=\sqrt{{\left(\gamma \cos\omega t\exp\left(-\frac{t}{T_{2}^{*}}\right)\right)}^{2}+{\left(\gamma \sin\omega t\exp\left(-\frac{t}{T_{2}^{*}}\right)\right)}^{2}}$$

For the matrix factorization method PSMF [47], positive-valued matrices are needed, and the original signal values must be converted to their logarithmic form for optimal analysis. To convert the absolute value in Equation (3) to a positive logarithmic form, the following Equation (4) (Appendix A) is applied:(4)$$V=\log_{10}\left(\left|z\right|+1\right)$$

Signal deconvolution can be then formulated as finding the factorization of the data matrix *V* (Appendix A):(5)$$V=W\cdot H+residuals=W_{signal}\cdot H_{signal}+W_{noise}\cdot H_{noise}+residuals$$

In this method using PSMF, we focus on sparse factorizations and on properly accounting for uncertainties while computing the factorization. Equation (5) estimates that the signal component ($W_{signal}\cdot H_{signal}$) decays exponentially with time, while the noise component ($W_{noise}\cdot H_{noise}$) is a random or flat value. To reconstruct the FIDs, the absolute value within each component is converted back to a complex value (Appendix A). The inverse STFT is computed by overlap-adding the inverse fast FT signals in each segment of the STFT spectrogram (Appendix A).

To evaluate SNR, both noise-removed and noise-only FIDs are converted to signal and noise spectra, respectively, by applying standard FT. SNR is calculated as the ratio of the signal peak intensity to the noise value by using the method of Mnova (Appendix A) [48]. The noise value is calculated by using the standard deviation of the signals-free region (Supplementary Equation (S17)). Finally, the relative SNR is the ratio of the SNR after denoising (*SNR_denoised_*) to the original SNR (*SNR_original_*), which is calculated as follows (Equation (6) and Appendix A):(6)$$Relative\;SNR=\frac{SNR_{denoised}}{SNR_{original}}$$

Figure 1 shows an example of application of our signal deconvolution process to sucrose ^1^H-NMR. STFT of the original FID adds a time axis to the frequency axis of the conventional FT spectrum (Figure 1a). The STFT spectrogram is three-dimensional, showing the frequency, time, and intensity of signal and noise. The matrix of the spectrogram was separated into signal and noise components based on the patterns of relaxation time using PSMF (Figure 1b). Each component was then converted into a signal FID and time-domain noise data by using inverse STFT (Figure 1c). Lastly, the time-region data were converted into the denoised spectrum and noise by using standard FT (Figure 1d). Regarding the noise reduction of the sucrose data, SNR of the denoised spectrum was improved about tenfold relative to the original data. In other words, for the sucrose sample, a 100-fold longer acquisition time would be required to obtain the same SNR without denoising. We compared signal and spectral quality between the original FT and noise reduction data (Appendix A). There was almost no difference between them.

In STFT, the size of a window function g(t−τ) is important. We define the percentage of the time width as the percentage of the window size to FID length. After examining different percentages of the time widths, we found that signal components could be properly extracted in 1.5% and 3.1% (512 and 1024 points for 33,280 points), but not 6.2% (2048 points for 33,280 points) (Appendix A). This is because the larger time width does not improve spectra since STFT becomes standard FT. Consequently, the percentage against the effective average region of FIDs is important for this method. Based on this result, the percentage of the time width was set to 3.1% for data analyzed in Figure 1. When using this method for data with short effective regions (fast relaxation systems such as solid-state NMR and quadrupole nucleus), data processing must be adjusted to maintain the shorter percentage of the time width. In addition, if an FID consists of a number of signals with differing *T*_2_*, it will not be possible to choose an optimal filter for all lines simultaneously by applying commonly used apodization. The apodization such as exponential filtering decreases both signal and noise. In contrast, the method that we propose enables signal and noise to be extracted from an FID based on each pattern of *T*_2_* relaxation time.

We compared the performance of PSMF with that of three other MF methods, namely standard nonnegative matrix factorization (NMF), sparse nonnegative matrix factorization (SNMF), and probabilistic nonnegative matrix factorization (PMF) (Figure 2). For PSMF, the noise region was successfully removed from the signal component (Figure 2a). For the other three methods, by contrast, the noise component remained in the signal component (Figure 2b–d). Regarding the PSMF time-varying coefficients, the signal component attenuated gradually over time, whereas the noise component attenuated sharply in the first segment and then became flat from the second segment (Appendix A). This observation suggests that part of the signal component may be included in the initial stage of the noise component. Therefore, for the optimal result in Figure 1, the initial value of the noise component is added as a signal component. The time-varying coefficients of the other three methods were characterized by containing mostly noisy components in the signal components, suggesting that the signal components were not properly extracted (Appendix A). The signal component is theoretically considered to be sparse data that comprise only specific frequency components. PSMF is a method that considers noise and uncertainty under the sparseness constraint, which suggests that it is suitable for removing noise from ^1^H-NMR data. We also examined the effect of the number of components in PMSF on signal deconvolution, which showed that it was possible to properly extract signal components when there were two components (Appendix A). When the number of components was increased, only noise components were separated more finely. Based on this result, the number of components was set to 2 in the signal deconvolution method for noise reduction. In the case of more complex data, such as the NMR signal of a mixture, it may be possible to apply the method to the characterization of multiple components by separating them with an arbitrary number of components.

### 2.2. Noise Reduction in NMR Data Measured by Various Pulse Sequences

The improvement in the relative SNR achieved by the noise reduction method was investigated by using large-scale data measured by various pulse sequences (Figure 3). Here, we analyzed the following three pulse sequences, which are generally used depending on the target of analysis: CPMG, which detects small molecules with long *T*_2_*, diffusion-edited, which detects proteins and lipids with relatively short *T*_2_*, and WATERGATE, which detects both of these. For the analysis of extensive data, percentages of the time width to FID lengths were set to 6.3% for CPMG and WATERGATE, 12.5% for diffusion-edited (1024 points for 16384 and 8192 points), and the initial three values of the noise component were added as a signal component. For CPMG and WATERGATE, the improvement rate was 3.7-fold and 3.3-fold, respectively. On the other hand, it was only 2.2-fold for diffusion-edited NMR data (Figure 3a). As a result of comparing the relative SNRs of three typical pulse sequences for 10 representative samples, the data of diffusion-edited tended to be lower than those of CPMG and WATERGATE as in the case of large-scale data (Figure 3b, Appendix A) since the time width for diffusion-edited (12.5%) is higher than that of the other two pulse sequences (6.3%). The SNR of any NMR data set is related to the acquisition parameters (Appendix A). In NMR data using CPMG and WATERGATE, the SNR is related to several acquisition parameters, such as receiver gain (RG), number of scans (NS), relaxation delay time (D1), spectral width (SW), and offset of the transmitter frequency (O1), whereas in diffusion-edited NMR, the SNR is particularly related to the gradient pulse in the *z*-axis (GPZ). In diffusion-edited NMR, signals from small molecules with long *T*_2_* relaxation times are suppressed. We therefore considered that, if the GPZ setting was insufficient, signals of small molecules would remain, resulting in a difference in relative SNR. As expressed, the peak SNR depends on *T*_2_* because an FID with large *T*_2_* yields a sharp line with higher SNR at the peak [38]. Thus, it seems likely that the diffusion-edited NMR data contain a lot of broad signals derived from macromolecules, resulting in less improvement as compared with CPMG and WATERGATE which have many sharp signals.

### 2.3. Application of Signal Deconvolution Method in Diffusion-Edited NMR

We further examined the application of our signal deconvolution method to diffusion-edited NMR data. For the optimal analysis of these data, the percentage of the time width to FID length was set to 6.3% (512 points for 8192 points), and the initial value of the noise component was added as a signal component. The original FID was separated into three components, including noise and the long and short components of *T*_2_* (Figure 4a,b). By extracting each component and performing standard FT, the SNR of the denoised spectrum was improved about threefold as compared with the original data. In addition, we obtained individual spectra for the short and long components of *T*_2_* (Figure 4c,d). Thus, the diffusion-edited spectrum was separated into signals from macromolecules and small molecules by the length of the *T*_2_* relaxation time. The composition of molecules in these signals is related to the GPZ value of the acquisition parameters (Appendix A). We consider that insufficient GPZ is the main factor affecting the relative SNR of diffusion-edited NMR data because, if GPZ is insufficient, relatively more signals from small molecules are contained in the measured signals. Knowing this composition will help to evaluate the data quality of diffusion-edited NMR.

### 2.4. Noise Factor Analysis in Data Measured by Low- and High-Field NMR at Multiple Institutions

To investigate the comprehensive relationship between noise and several acquisition parameters, we analyzed noise factors in data acquired by low- and high-field NMR at multiple institutions. We collected NMR data for four compounds (glucose, sucrose, citric acid, and lactic acid) measured by benchtop NMR (60 MHz) and high-field NMR (500, 600, and 700 MHz) from five institutions/data repositories (RIKEN, NUIS (Niigata University of International and Information Studies), BMRB [49], BML [50], and HMDB [51]) (Appendix A). The results of correlation analysis between noise and experimental parameters were first summarized as a heatmap (Appendix A). With a specific focus on the experimental parameters that affect the SNR, we then derived a network of experimental factors affecting noise based on the correlation coefficients between SNR and experimental parameters (Figure 5). Here, in addition to the SNR calculated using Mnova, we calculated a theoretical SNR value (calcSNR) using a previously described SNR formula (Appendix A) [52] in order to obtain a theoretical SNR index based on acquisition parameters. Figure 5 shows that, based on the correlation between SNR and, for example, number of scans (NS) and signal intensity (e.g., standard, sample, and solvent), the integration of strong signals will increase noise and reduce SNR. Therefore, the suppression of water signals and sample concentration will be important factors to obtain NMR data with a good SNR.

In situations where longer NMR measurements are needed owing to poor signals (e.g., for nuclei of low sensitivity and/or low natural abundance, and samples of low concentration), paying attention to the certain factors, as discussed here, may provide significant improvements in SNR [38], or even more marked savings in measurement time for a given SNR. For example, too long an acquisition time is not beneficial for SNR. An FID of the time constant *T*_2_* gives, on Fourier transformation, a line width of 1/π*T*_2_* or approximately 1/3*T*_2_*. Thus, data acquisition beyond about 3*T*_2_* provides little gain in resolution, but causes a considerable deterioration in SNR. In addition, the spectral width may be set high enough to prevent aliasing of NMR signals. If not, there may be still other signals that fold, namely noise, meaning that the final SNR in the spectrum deteriorates.

Receiving efficiency (*R*) has been proposed as a way to characterize how efficiently the NMR signal can be observed after a unit transverse magnetization in a sample under optimal probe tuning and matching conditions [39]. In that study, the NMR signal amplitude was described as a function of the instrument constant, receiver gain, excitation angle *θ*, inhomogeneity factor *I*(*θ*), concentration of the observed nucleus, and sample volume. Modern NMR spectrometers require receivers to work within their linear ranges to maintain high-fidelity line shapes and peak integration [40]. The NMR receiver gain is a parameter that is often chosen to maximize SNR. For example, for optimal sensitivity, a dilute analyte needs to be observed with high NMR receiver gain, while the strong, interfering solvent signal must be suppressed [41]. In this case, the dependence of *I*(*θ*) on *θ* becomes more significant because homogeneity is typically lower for a cryoprobe than for its conventional counterpart [42], and failing to recognize the dependence of *I*(*θ*) on *θ* alone may potentially lead to errors in quantification as large as 5%. Other factors that we have discussed have less effect on SNR, but are significant in terms of line shape.

## 3. Materials and Methods

### 3.1. Signal Deconvolution Method

The signal deconvolution method was developed in python 3, and built as a graphical user interface (GUI) tool using Tkinter. The tool is available on http://dmar.riken.jp/NMRinformatics/. The processing of NMR data was implemented by using the nmrglue [53] package in Python. PSMF [47], PMF [54], SNMF [55], and standard NMF [56] were calculated based on the NIMFA Python library for nonnegative matrix factorization [44].

### 3.2. Noise Factor Analysis Method

The noise factor analysis consisted of four steps implemented in python 3, namely: (1) Collecting acquisition parameters of NMR data: FID and acquisition parameters were searched from the selected NMR data directory and written to CSV files. (2) Calculating SNR: each FID was usually processed to an FT spectrum and denoised spectrum, and the SNR and its improvement ratio were calculated. In the noise factor analysis of data collected from multiple databases, SNR was calculated by using Mnova. (3) Calculating the correlation coefficient between SNR and each parameter by Pearson’s correlation coefficient. (4) Visualizing experimental factors: the nodes, edges, and widths of networks based on the correlation coefficient were transformed in GML format by using the Networkx package in Python. Lastly, the network figure was drawn by using Cytoscape [57].

### 3.3. NMR Data Acquisition

Briefly, ^1^H-NMR data were by recorded using an Avance II 700 Bruker spectrometer equipped with a 5-mm inverse CryoProbe operating at 700.153 MHz for ^1^H. In the ^1^H -NMR data, the number of data using CPMG pulse sequence was 2386, the number of data using WATERGATE pulse sequence was 2760, and the number of data in the 1D LED experiment using bipolar gradients (diffusion-edited) pulse sequence was 975 [58,59,60,61]. Regarding these large data sets, a summary of information on the sample and acquisition parameters (the sample title, solvent, acquisition time, acquisition point, and the original SNR) is available at http://dmar.riken.jp/NMRinformatics/. Data sets for comparing the relative SNRs of three typical pulse sequences for 10 representative samples are shown in Appendix A. To demonstrate the denoising method, data for sucrose and citric acid were acquired by using the presaturation (program name; “zgpr”) pulse sequence. To demonstrate the method of separating signals in the diffusion-edited spectrum, ^1^H-NMR data for fish muscle were measured by a diffusion-edited pulse sequence. Lastly, 48 sets of ^1^H-NMR data (glucose, sucrose, citric acid, and lactic acid) were collected from the following five sites; RIKEN, NUIS, BMRB, BML, and HMDB. The data were measured with NMR spectrometers of 60, 500, 600, and 700 MHz manufactured by Bruker, Varian, and Nanalysis (Appendix A).

## 4. Conclusions

We have developed a measurement informatics tool for NMR signal deconvolution and noise factor analysis and used it to investigate the relationship between noise and acquisition parameters in accumulated NMR datasets. This method enables 1D-NMR spectra to be evaluated with a high SNR, and residual signals from small molecules to be removed from diffusion-edited spectra. This method can be adjustable to any *T*_2_* length, recycle delay, sample molecular weight, or measurement temperature. The percentage of the time width against the effective average signal region of FIDs must be adjusted according to *T*_2_* length. Therefore, when using this method for fast relaxation systems such as solid-state NMR and quadrupole nucleus, additional efforts are needed. In the case of 2D-NMR, it is necessary to use this method by splitting each *t*_1_-dimensional FID and creating a series of sub-FIDs. Noise factor analysis of accumulated NMR datasets might facilitate the investigation of experimental factors related to a lower SNR. Therefore, these methods will help to determine optimal acquisition parameters, to cleanse data, including data management and noise reduction in accumulated NMR datasets, and to promote data-driven studies of molecular complexity using NMR.

## Figures and Tables

**Figure 1 ijms-21-02978-f001:**
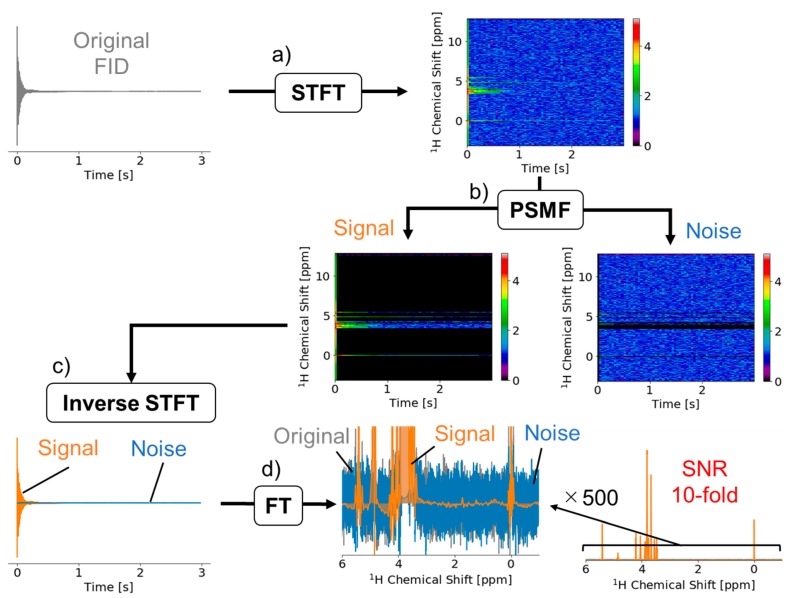
The free induction decay (FID) signal deconvolution method and its application to ^1^H-NMR data for sucrose. (**a**) The spectrogram was obtained by applying short-time Fourier transform (STFT) to the original FID. (**b**) The matrix obtained after STFT was applied to probabilistic sparse matrix factorization (PSMF), which separated it into signal and noise components. (**c**) The signal and noise components were converted into a noise-removed FID signal (orange) and a time-domain noise signal (blue) by using inverse short-time Fourier transform. (**d**) Finally, the noise-removed FID and the time-domain noise signal were converted to a frequency-domain spectrum by applying standard Fourier transform. As compared with the original FID, the signal-to-noise ratio of the denoised FID was improved about tenfold.

**Figure 2 ijms-21-02978-f002:**
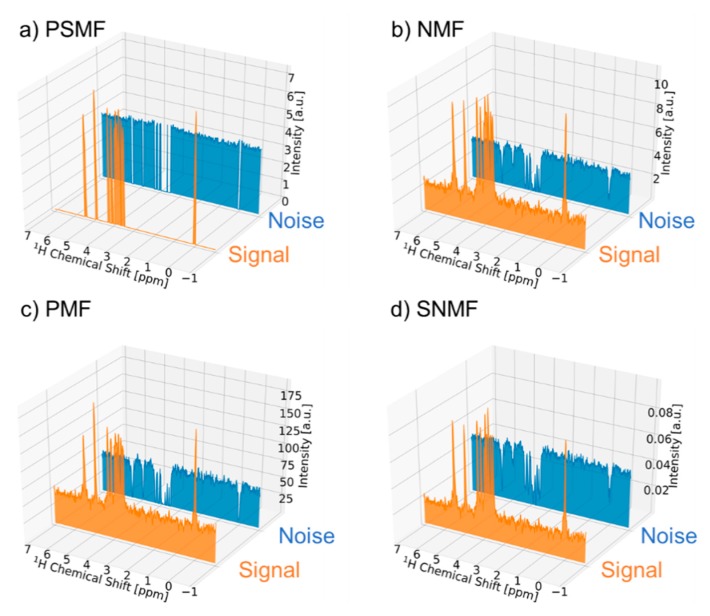
Comparison of four matrix factorization (MF) methods in signal deconvolution. Shown are spectral patterns of signal deconvolution for sucrose ^1^H-NMR data using (**a**) PSMF, (**b**) NMF, (**c**) PMF, and (**d**) SNMF. The signal components are shown in orange and the noise components are shown in blue.

**Figure 3 ijms-21-02978-f003:**
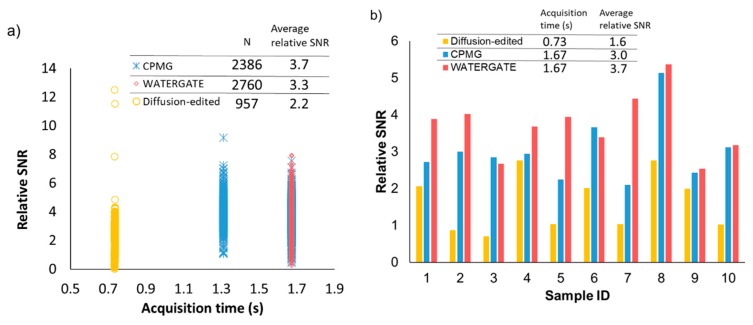
Relative SNR in data measured by three pulse sequences. (**a**) Shown is the relationship between the relative SNR after application of the noise reduction method to large-scale data measured by three pulse sequences: CPMG (blue), WATERGATE (red), and diffusion-edited (yellow), and its acquisition time. The upper part of the figure shows the number of spectra and the average relative SNR for each pulse sequence. (**b**) Comparison of the efficiency for improvement of the SNR measured by three pulse sequences: CPMG (blue), WATERGATE (red), and diffusion-edited (yellow), among NMR spectra derived from sample ID of 1 to 10. The acquisition time and the average relative SNR for each pulse sequence are shown in the upper part of the figure.

**Figure 4 ijms-21-02978-f004:**
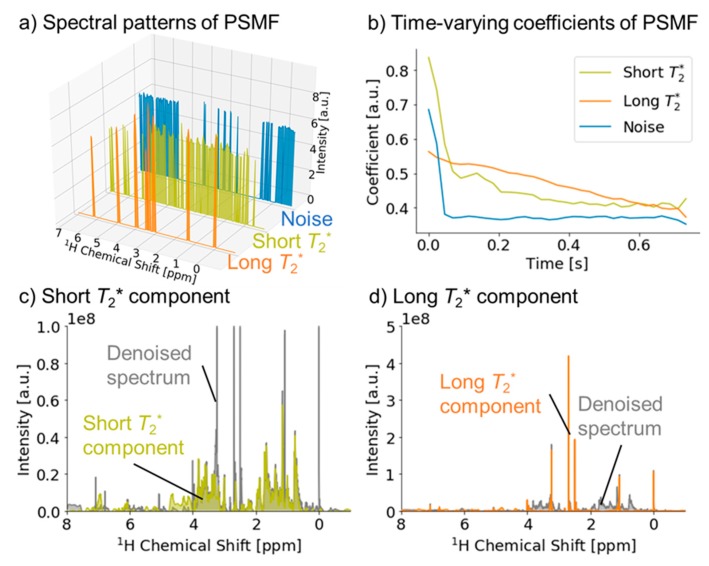
Application of the signal deconvolution method to diffusion-edited spectra. (**a**) Spectral patterns showing signals from small molecules (orange) and macromolecules (olive) separated by the length of the *T*_2_* relaxation time, and noise (blue). (**b**) Time-varying coefficients of each component in MF. (**c**) Denoised spectrum (gray), and spectrum of the short *T*_2_* component (olive). (**d**) Denoised spectrum (gray), and spectrum of the long *T*_2_* component (orange).

**Figure 5 ijms-21-02978-f005:**
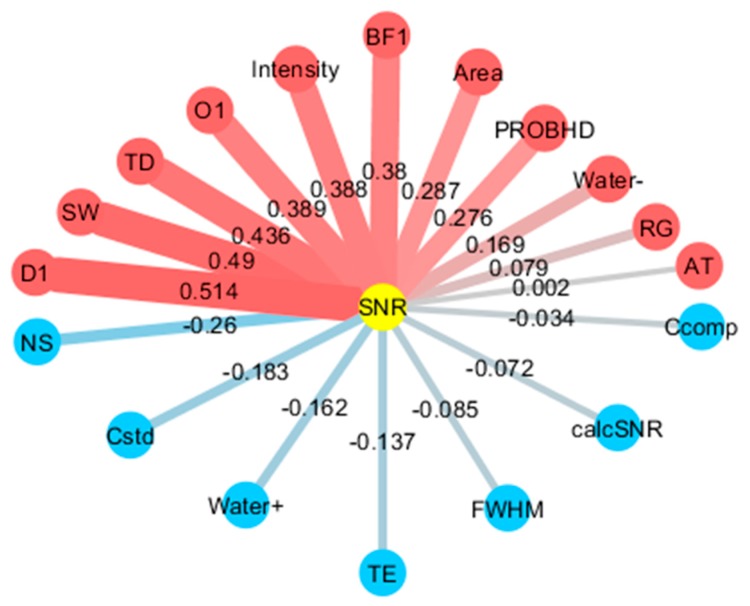
Analysis of experimental factors based on a correlation network of SNR and experimental parameters. The network diagram was drawn by setting positive correlations to red, negative correlations to blue, and the magnitude of the correlation coefficient to the edge thickness. Abbreviations: SNR, signal-to-noise ratio; calcSNR, calculated SNR; Cstd, concentration of standard compound; Ccomp, concentration of compound; Water+, positive intensity of water signal peak to standard peak; Water–, negative intensity of water signal peak to standard peak; Intensity, intensity of standard signal; FWHM, full width at half maximum; Area, area of standard signal; RG, receiver gain; NS, number of scans; D1, relaxation delay time; SW, spectral width; AT, acquisition time; TD, time-domain data size; O1, offset of transmitter frequency; TE, temperature; BF1, basic transmitter frequency for channel F1 in Hertz; PROBHD, if cryoprobe, value is 4, if not, value is 0.

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
