# Peer review of "Signal Deconvolution and Noise Factor Analysis Based on a Combination of Time–Frequency Analysis and Probabilistic Sparse Matrix Factorization"

_ijms, 2020, doi:10.3390/ijms21082978_

Round 1

Reviewer 1 Report

Ijms-770925 by Shunji Yamada et al.
This paper discusses mathematical approach of noise reduction for NMR FID signals to produce Fourie transformed spectra. The objective is very important to improve the signalto- noise ratio without any hardware upgrade for NMR research field. I think this paper may be valuable to be published in this International Journal of Molecular Sciences, but some ambiguities should be revised listed below before acceptance.
1. Page 4, line 159: Authors insist the importance of time width for STFT. However, the provide time width is point number not time. I wonder the points does not show correct time length because of variety of dwell time. It is necessary to explain the time length. By the way, what is a time width? Is it a window function g(t-tau)’s length? It should be explained politely.
Furthermore, I think the effect of the time width depends on the length of FID. If one FID has 8k points and another 32k, the effect of 512, 1024, or 2048 points for time width on STFT becomes different; off course, the effective signal (not noise) FID length is much more important. Actually, the STFT with 2048 points of time width in Figure S2 shows less noise reduction. This is because 2048 points are close to the effective signal points in an FID, so it is reasonable that the large points time width does not work well, and STFT becomes normal FT. Consequently, I think that important issue is not points but the percentage against the effective signal FID length.

2. Page 6, line 205, “section 2.2”: Figure 3 shows the various data obtained various samples using three pulse sequences. However, the colors are not three but five. Could you explain it much precisely?
The total number of spectra is appeared in page 10 as 2386 for CPMG, 2761 for WATERGATE, and 956 for diffusion-edited, namely 6103 spectra. However, there is no information about samples. I think the noise reduction effect depends on kinds of sample (concentration, solvent, and so on) and acquisition time, acquired points, and original signal-to-noise ratio; although there is some comments about it in the supplemental file.
This section and Figure 3 are not meaningful. For example, page 7 line 226, “a long acquisition time leads to the accumulation of an excessive amount of noise and thus decreases SNR in the transformed spectrum.”, the readers can not recognize which data in Figure 3 are observed with longer acquisition time: how or which data should be compared. Moreover, it is necessary to compare the efficiency for improvement of the signal-to-noise ratio among three CPMG, WATERGATE, and diffusion-edited pulse sequences using the same sample. Which data in Figure 3 are you obtained from the same sample? The readers never understand the Figure 3 correctly. Much precise and polite description is required in this section, otherwise this section is not necessary.Discussion is very poor.

3. Are there any limitations to apply this method, for example the T2* length, recycle delay, molecular weight of sample, multi-nuclei, measurement temperature, and so on? It should be discussed.
4. It should be compared spectral quality, such as J coupling appearance, quantitatively for signal intensity, and peak shape between the original FT and the noise reduction method.
This Journal is not mathematical one but about molecular science.

Author Response

This paper discusses mathematical approach of noise reduction for NMR FID signals to produce Fourie transformed spectra. The objective is very important to improve the signalto- noise ratio without any hardware upgrade for NMR research field. I think this paper may be valuable to be published in this International Journal of Molecular Sciences, but some ambiguities should be revised listed below before acceptance.

[Response]

Thanks for your understanding and appropriate comments on the mathematical approach to noise reduction of NMR FID signals. We considered responses to all comments as follows.

  1. Page 4, line 159: Authors insist the importance of time width for STFT. However, the provide time width is point number not time. I wonder the points does not show correct time length because of variety of dwell time. It is necessary to explain the time length. By the way, what is a time width? Is it a window function g(t-tau)’s length? It should be explained politely.

Furthermore, I think the effect of the time width depends on the length of FID. If one FID has 8k points and another 32k, the effect of 512, 1024, or 2048 points for time width on STFT becomes different; off course, the effective signal (not noise) FID length is much more important. Actually, the STFT with 2048 points of time width in Figure S2 shows less noise reduction. This is because 2048 points are close to the effective signal points in an FID, so it is reasonable that the large points time width does not work well, and STFT becomes normal FT. Consequently, I think that important issue is not points but the percentage against the effective signal FID length.

[Response]

The time width is the length of the window function . We added the following explanation on line 180 for the percentage of to the time width to FID length: Percentage of the time width (a window function ’s length) to FID length is important in STFT. After examining different percentages of the time widths to FID length, we found that signal components could be properly extracted in 1.5% and 3.1% (512 and 1024 points for 33280 points), but not extracted at 6.2% (2048 points for 33280 points).

  1. Page 6, line 205, “section 2.2”: Figure 3 shows the various data obtained various samples using three pulse sequences. However, the colors are not three but five. Could you explain it much precisely?

The total number of spectra is appeared in page 10 as 2386 for CPMG, 2761 for WATERGATE, and 956 for diffusion-edited, namely 6103 spectra. However, there is no information about samples. I think the noise reduction effect depends on kinds of sample (concentration, solvent, and so on) and acquisition time, acquired points, and original signal-to-noise ratio; although there is some comments about it in the supplemental file.

This section and Figure 3 are not meaningful. For example, page 7 line 226, “a long acquisition time leads to the accumulation of an excessive amount of noise and thus decreases SNR in the transformed spectrum.”, the readers can not recognize which data in Figure 3 are observed with longer acquisition time: how or which data should be compared. Moreover, it is necessary to compare the efficiency for improvement of the signal-to-noise ratio among three CPMG, WATERGATE, and diffusion-edited pulse sequences using the same sample. Which data in Figure 3 are you obtained from the same sample? The readers never understand the Figure 3 correctly. Much precise and polite description is required in this section, otherwise this section is not necessary.Discussion is very poor.

[Response]

We have shown the relative SNR with acquisition time for large data of three pulse sequences to show the efficiency to improve the signal-to-noise ratio (Fig. 3a). For these large data sets, a csv file that summarizes the sample title, solvent and acquisition time, acquisition point, and the original signal-to-noise ratio as information about the sample and acquisition parameters is available at http://dmar.riken.jp/NMRinformatics/SIforDCTN.zip. Furthermore, Comparison of the efficiency for improvement of the SNR measured by three pulse sequences: CPMG (blue), WATERGATE (red), and diffusion-edited (yellow), among same NMR spectra derived from sample ID of 1 to 10. The points described above are added on line 241.

  1. Are there any limitations to apply this method, for example the T2* length, recycle delay, molecular weight of sample, multi-nuclei, measurement temperature, and so on? It should be discussed.

[Response]

In this paper, we focused on 1D-NMR for applying our signal separation method. This technique has in principle no restrictions on T2 * length, recycle delay, sample molecular weight, or measurement temperature. However, when using this method for fast relaxation systems such as solid-state NMR and quadrupole nucleus, additional efforts are needed to handle due to short relaxation times. Furthermore, in the case of two-dimensional NMR, it is necessary to employ this method by splitting each t1-dimensional FID and creating a series of sub-FIDs. The above points are added at line 371.

  1. It should be compared spectral quality, such as J coupling appearance, quantitatively for signal intensity, and peak shape between the original FT and the noise reduction method.

This Journal is not mathematical one but about molecular science.

[Response]

We compared the signal quality and the spectra quality in 1H-NMR data of citric acid between the original FT and the noise reduction method. The results were shown in Figure S2 and Table S1 as supplemental material. The points described above are added on line 277.

Reviewer 2 Report

Subject of the paper is very interesting for NMR users. Authors suggested a new approach (matrix factorization) to reduce noise of FID signal, increase signal-to-noise ratio and to deconvolute the FID signal.

I have few questions and comments to the authors.

  1. There is a review of the problem where many methods are referred. However, no references on Piter Stilbs computational approaches like CORE, which were successfully applied to solve the same problems of NMR spectroscopy related with signal-to-noise ratio and deconvolution, particularly for NMR relaxometry and NMR diffusometry.
  2. Stilbs, K. Paulsen, P.C. Griffiths, J. Phys. Chem. 1996, 100, 8180.
  3. Stilbs, "Automated CORE, RECORD, and GRECORD processing of multi-component PGSE NMR diffusometry data," European Biophysics Journal, vol. 42, no. 1, pp. 25-32, 2013.
  4. Stilbs, "RECORD processing : A robust pathway to component-resolved HR-PGSE NMR diffusometry," Journal of magnetic resonance, vol. 207, no. 2, pp. 332-336, 2010.
  5. What is the advantage of the methods in comparison with previously used methods?
  6. Line 48. What means the term “molecular resolution”?
  7. Line 65. “because of the effect of magnetic field homogeneity” should be “because of the effect of magnetic field inhomogeneity”
  8. Line 74. What is the “PCA”?
  9. Line 233. “Figure 7”. There is no Figure 7 there.

Author Response

Comments and Suggestions for Authors

Subject of the paper is very interesting for NMR users. Authors suggested a new approach (matrix factorization) to reduce noise of FID signal, increase signal-to-noise ratio and to deconvolute the FID signal.

I have few questions and comments to the authors.

[Response]

Thanks for your understanding and appropriate comments on our signal separation method. We considered responses to all comments as follows.

There is a review of the problem where many methods are referred. However, no references on Piter Stilbs computational approaches like CORE, which were successfully applied to solve the same problems of NMR spectroscopy related with signal-to-noise ratio and deconvolution, particularly for NMR relaxometry and NMR diffusometry.

Stilbs, K. Paulsen, P.C. Griffiths, J. Phys. Chem. 1996, 100, 8180.

Stilbs, "Automated CORE, RECORD, and GRECORD processing of multi-component PGSE NMR diffusometry data," European Biophysics Journal, vol. 42, no. 1, pp. 25-32, 2013.

Stilbs, "RECORD processing : A robust pathway to component-resolved HR-PGSE NMR diffusometry," Journal of magnetic resonance, vol. 207, no. 2, pp. 332-336, 2010.

[Response]

We agreed that computational approach papers such as CORE, written by Piter Stilbs et al., provided useful information for our relaxation time focused signal separation approach. We quoted them on line 80.

What is the advantage of the methods in comparison with previously used methods?

[Response]

Previously reported signal separation methods require multiple FIDs measured in multiple dimensions or multiple parameters. Furthermore, in order to recognize a signal, it is necessary to set a reference spectral pattern and a region of interest. On the other hand, our proposed method of signal separation in this article can generate multiple spectra for each time by short-time Fourier transformation and it provides a signal pattern by relaxation time, so it can be performed with only one FID of 1D-NMR. Furthermore, a GUI tool (http://dmar.riken.jp/NMRinformatics/) written in Python that automatically separates signals for individual data following FID data search has been developed, so no programming knowledge is required for the users. A computational science approach that focuses on diffusion coefficients in PFG-NMR, such as CORE, requires a specific NMR probe with a coil wound to generate a pulsed magnetic field gradient (PFG). On the other hand, our method focusing on relaxation time utilizes the attenuation behavior of the FID signal, which is the basis of the NMR method, and can be used regardless of hardware, and it is applicable to spectroscopy as well as relaxometry / diffusometry. Thus, differences in molecular composition in mixture sample can be evaluated. We have added to line 95 these advantages of the methods in comparison with previously published methods.

Line 48. What means the term “molecular resolution”?

[Response]

In line 48, we are discussing the problem of signal overlap in the spectrum and corrected it for the appropriate "spectral resolution" rather than the "molecular resolution".

Line 65. “because of the effect of magnetic field homogeneity” should be “because of the effect of magnetic field inhomogeneity”

[Response]

In line 65, we are discussing the effects of magnetic field inhomogeneity on the relationship between T2 and T2 *, and corrected the sentence to "because of the effect of magnetic field inhomogeneity".

Line 74. What is the “PCA”?

[Response]

In line 74, “PCA” is Principal Component Analysis, one of the most commonly used multivariate analysis methods. We have added "Principal Component Analysis (PCA), one of the most commonly used multivariate analysis methods to extract data features" to the text.

Line 233. “Figure 7”. There is no Figure 7 there.

[Response]

In line 233, we described "Figure S7". However, we corrected this section to emphasize Figure 4. Therefore, Figure S7 was shown as a supplement.